# Development of Loop-Mediated Isothermal Amplification Assay for Rapid Detection of *Cucurbit Leaf Crumple Virus*

**DOI:** 10.3390/ijms21051756

**Published:** 2020-03-04

**Authors:** Sumyya Waliullah, Kai-Shu Ling, Elizabeth J. Cieniewicz, Jonathan E. Oliver, Pingsheng Ji, Md Emran Ali

**Affiliations:** 1Department of Plant Pathology, University of Georgia, Tifton, GA 31793, USA; Sumyya.Waliullah@uga.edu (S.W.); jonathanoliver@uga.edu (J.E.O.); pji@uga.edu (P.J.); 2U. S. Vegetable Laboratory, USDA-ARS, Charleston, SC 29414, USA; kai.ling@usda.gov; 3Department of Plant and Environmental Sciences, Clemson University, SC 29634-0310, USA; ecienie@clemson.edu

**Keywords:** *Cucurbit leaf crumple virus*, begomovirus, detection, loop-mediated isothermal amplification

## Abstract

A loop-mediated isothermal amplification (LAMP) assay was developed for simple, rapid and efficient detection of *Cucurbit leaf crumple virus* (CuLCrV), one of the most important begomoviruses that infects cucurbits worldwide. A set of six specific primers targeting a total 240 nt sequence regions in the DNA A of CuLCrV were designed and synthesized for detection of CuLCrV from infected leaf tissues using real-time LAMP amplification with the Genie^®^ III system, which was further confirmed by gel electrophoresis and SYBR™ Green I DNA staining for visual observation. The optimum reaction temperature and time were determined, and no cross-reactivity was seen with other begomoviruses. The LAMP assay could amplify CuLCrV from a mixed virus assay. The sensitivity assay demonstrated that the LAMP reaction was more sensitive than conventional PCR, but less sensitive than qPCR. However, it was simpler and faster than the other assays evaluated. The LAMP assay also amplified CuLCrV-infected symptomatic and asymptomatic samples more efficiently than PCR. Successful LAMP amplification was observed in mixed virus-infected field samples. This simple, rapid, and sensitive method has the capacity to detect CuLCrV in samples collected in the field and is therefore suitable for early detection of the disease to reduce the risk of epidemics.

## 1. Introduction

*Cucurbit leaf crumple virus* (CuLCrV) is a single-stranded DNA virus belonging to the Begomovirus genus within the Geminiviridae family. The various biotypes of whitefly *Bemisia tabaci* (Genn.), including the silverleaf whitefly (*B. tabaci* biotype B = *B. argentifolii* Bellows & Perring), transmit CuLCrV in a persistent and circulative manner [1]. CuLCrV is able to infect most cucurbits, including cucumber, cantaloupe, squash (yellow, zucchini, and winter squash), pumpkin, and watermelon [1,2] and has been reported to infect bean plants [3]. CuLCrV has been identified in several states of the United States, including Arizona, Texas, California [2,4], Florida [1,5], South Carolina [6] and Georgia [7], as well as in north-central Mexico [4]. Georgia and Florida are major cucurbit (squash, watermelon, pumpkin, cucumber and cantaloupe) producing states [8]. More than 40% of cucurbit production occurs during the fall in these states, when the plants are most threatened by whiteflies and a complex of whitefly-transmitted viruses. Among those viruses, CuLCrV has caused severe damage to cucurbits and snap beans along with other whitefly-transmitted viruses in recent years [9,10]. In 2016 and 2017, whitefly-transmitted CuLCrV and other viruses in Florida and Georgia affected hundreds of fields spanning more than 10,000 acres, with over $50 million lost in Georgia alone [10]. Symptoms of CuLCrV include yellow chlorotic spots (abnormally pale due to insufficient chlorophyll), interveinal yellowing, mosaic, and leaf curling and crumpling [2,11]. In the case of severe infection, stunting and growth distortion are observed [11]. These symptoms can also resemble those caused by other closely related whitefly-transmitted begomoviruses such as *Squash leaf curl virus* (SLCV), *Squash mild leaf curl virus* (SMLCV), *Bean calico mosaic virus* (BCaMV) and *Melon chlorotic leaf curl virus* (MCLCV). SLCV also causes severe leaf chlorosis, leaf crumpling, curl symptoms and stunting of squash and melon plants, and it cannot be easily distinguished from CuLCrV on the basis of symptoms alone [11]. Therefore, proper identification requires molecular or serological methods. 

Several methods have been used to detect and distinguish CuLCrV from other begomoviruses. Among those, polymerase chain reaction (PCR) is the most commonly used molecular method [5,11,12,13]. Degenerate primers are also widely used for PCR amplification to distinguish the identity of the virus from other begomoviruses [3,5,6,12,14]. Idris et al. [15] described the detection of CuLCrV and other begomoviruses with RFLP following PCR amplification using degenerate primers and digestion with restriction enzymes AccI, SalI, or SpeI to confirm the identity of the viral DNA-A. In another study, Turechek et al. [16] used nucleic acid hybridization after blotting the cross-sections of symptomatic watermelon plants onto a nitrocellulose membrane to detect CuLCrV. Typically, serological methods like ELISA are more practical and cheaper than other methods. However, the detection procedures are more labor-intensive and carry the risk of non-specific binding, which can give false-positive results. Also, ELISA requires centralized laboratory equipment, relatively high sample volume, and has relatively low sensitivity [17,18]. By contrast, nucleic acid hybridizations have the advantage of detecting a specific gene or nucleic-acid sequence, so gene expression is not a prerequisite. Unfortunately, these methods are tedious and more sophisticated and complex than traditional methodologies [19,20,21]. PCR and qPCR assays have limitations in pathogen identification because of the need for specialized laboratory equipment and reagents, as well as proper training and technical expertise, which are often unavailable in poorly-resourced laboratory and field conditions [22,23,24].

Loop-mediated isothermal amplification (LAMP) assay is a novel technique that can effectively address the limitations of the methods described above. It is a straightforward, rapid, highly sensitive, specific, cost-effective method that can be used for early diagnosis and in-situ testing of crop pathogens [25,26,27]. LAMP is based on auto-cycling strand displacement DNA synthesis by a Bst (*Bacillus stearothermophilus*) DNA polymerase utilizing four to six primers, which bind laterally to the distinct sites for highly specific amplification in isothermal conditions [26,27,28,29,30,31]. The amplified product from LAMP can be detected using gel electrophoresis, or by adding an intercalating dye (mostly with SYBR™ Green I) to the final end product for visual inspection, or using real-time quantitative measurement by either measuring turbidity using turbidimeter or fluorescence using Genie^®^ II or Genie^®^ III (OptiGene, Horsham, WS, UK) [31,32,33,34]. These factors make a LAMP assay suitable to use in laboratory or field conditions.

To diagnose several important DNA and RNA plant viruses, LAMP and reverse transcription LAMP (RT-LAMP) have been used. These viruses include *Squash leaf curl virus* (SLCV) [35], *Tomato chlorosis virus* (ToCV) [36], *Cucurbit chlorotic yellows virus* (CCYV) [37,38], *Tomato yellow leaf curl virus* (TYLCV) [39], *Tomato spotted wilt virus* (TSWV) [40], *Cucumber mosaic virus* (CMV) [41], *Potato virus Y* (PYV) [42], *Potato leafroll virus* (PRLV) [43], *Cucumber green mottle mosaic virus* (CGMMV) [44], *Chinese wheat mosaic virus* (WCHMV0) [45], *Barley yellow dwarf viruses* (BYDV) [46], *Japanese yam mosaic virus* (JYMV) [47], *Plum pox virus* (PPV) [48], *Tobacco mosaic virus* (TMV) [49], nine rice-infecting viruses [50], and three begomoviruses infecting tomato in Panama, i.e., *Potato yellow mosaic Panama virus* (PYMPV), *Tomato leaf curl Sinaloa virus* (STLCV) and *Tomato yellow mottle virus* (TYMV) [51]. However, no LAMP technique for the identification of CuLCrV has been previously reported. The objective of this study was to develop a LAMP assay for rapid and efficient detection of CuLCrV in cucurbits. This assay should supplement and enhance existing procedures for detecting the pathogen.

## 2. Results

### 2.1. Optimization of LAMP Conditions for CuLCrV Detection

The optimal temperature and reaction time for the LAMP assay were determined for CuLCrV detection from infected samples. When the gradient LAMP reaction was performed from 66 to 73°C using 1.0 × 10^−1^ ng/µL of CuLCrV-infected diluted DNA sample, it was revealed that the reaction conducted at 71°C had the optimal outcome with a shortest peak time (Ti_amp_) of 18 min and 15 sec and a melting curve with specific peak at 86.58 °C (Appendix A, Appendix A). No amplification was found at lower than 65 °C using LavaLAMP™ DNA Master Mix (Lucigen, Middleton, WI, USA) (Appendix A). Next, the LAMP reaction was performed at 71 °C for 30, 45, 60 or 75 min intervals with 5.0 × 10^−4^ ng/µL of CuLCrV-infected DNA sample. No amplification was observed from the 30 and 45 min reaction intervals, but a peak amplification time of 46 min 15 sec was observed in the longer (60 and 75 min) reactions. No visible difference was observed for gel electrophoresis and with SYBR™ Green 1 nucleic acid gel stain (Invitrogen, Carlsbad, CA, USA) for both time cycles for the amplified product (data not shown). However, to provide adequate lengths of time, 60 min was determined to be optimal reaction time to complete amplification of CuLCrV from infected samples with a low concentration of infection by LAMP assay based on the findings from this study and previously reported studies [35,38,52]. Therefore, the optimal reaction conditions for CuLCrV-LAMP were determined to be 71 °C for 60 min.

### 2.2. Evaluation of LAMP Primers with CuLCrV-Infected Samples

LAMP primer sets including F3, B3, FIP [F1c-F2], BIP [B1c-B2], LF and LB designed by Primer Explorer version 5 (Figure 1, Appendix A) could detect CuLCrV from infected samples collected from several different hosts, including yellow squash, zucchini, watermelon and cucumber (Figure 2). The LAMP assay could detect CuLCrV from infected samples by gel electrophoresis (Figure 2A), visual inspection with SYBR™ Green 1 (Invitrogen, Carlsbad, CA) nucleic acid gel staining (Figure 2B), and real-time amplification with Genie^®^ III (OptiGene, Horsham, WS, UK) (Figure 2C). However, the detection variability among samples by LAMP (Figure 2A–C) indicated a variable titer of virus infection among the infected samples, which was further quantified by qPCR (Figure 2D).

### 2.3. Specificity of LAMP Detection of CuLCrV

Total DNA extracted from CuLCrV, SLCV, SMLCV -infected squash, TYLCV and cDNA of TSWV infected tomato and DNA of healthy squash leaf tissue as negative control were evaluated to determine the specificity of the LAMP method (Figure 3). Following amplification using 50 ng each of nucleic acids, only the CuLCrV-infected sample could be detected by the characteristic ladder-like pattern in the agarose gel, while others remained undetected (Figure 3A). The results obtained from SYBR™ Green I gel staining and Genie^®^ III real-time amplification was consistent with the gel-based analysis. Only the CuLCrV-infected sample resulted in a florescent bright green color using SYBR™ Green I gel staining compared to the orange color of the failed reaction (Figure 3B) and an amplified curve by Genie^®^ III in contrast to a non-amplified curve of the other samples (Figure 3C). The absence of visible amplification of DNA in the healthy control or other virus-infected samples by LAMP indicated that the primer was specific for the detection of CuLCrV without cross-reactivity with other closely related viruses (Figure 3). 

To check whether the CuLCrV LAMP amplification was due to the low titer of other viruses, qPCR was performed using virus-specific primers for virus quantitation (Appendix A). Except SMLCV, all other virus titers were either higher or close to the titer of CuLCrV (Figure 3D). Therefore, the observed specificity of the LAMP amplification was not due to the low titer of other viruses in comparison with CuLCrV DNA.

To further confirm the specificity of the LAMP primers for amplification of CuLCrV and to check whether the primer efficiency was reduced in a mixed virus assay, five different DNA template mixture were prepared as described in the method section. Successful LAMP amplification was observed in the sample prepared to include both CuLCrV and other mixed virus DNA (Figure 4). In gel electrophoresis, no visible difference was observed between the LAMP product amplified from CuLCrV DNA mixed with healthy control versus the product amplified from CuLCrV DNA with the other mixed virus DNA (Figure 4B). This was also demonstrated using real-time amplification with Genie^®^ III (Figure 4C). In addition, the amplification of only CuLCrV from the mixed virus assay was confirmed by CuLCrV-specific PCR and direct sequencing of the PCR product (Figure 4A). Furthermore, no amplification was observed from the mixed virus DNA sample, which did not include CuLCrV or in the healthy control or negative control (Figure 4). Thus, the developed LAMP assay specifically amplified only CuLCrV.

### 2.4. Comparative Sensitivity Analysis of LAMP Compared to PCR and qPCR

A series of 5-fold dilutions of total DNA (10 to 10^−5^ ng/μL) extracted from a CuLCrV-infected squash leaf sample was used to compare the sensitivity of LAMP, PCR and qPCR assay (Figure 5). The results demonstrated that DNA concentration of 5.0 × 10^−3^ ng/µL (5.0 pg/µL) was detectable by PCR, although at this low level, the amplified product was hardly visible and the resulting band at the DNA concentration of 1.0 × 10^−2^ ng/µL (10.0 pg/µL) was clear and much brighter. By contrast, dilutions up to 5.0 × 10^−4^ ng/µL (0.5 pg/µL) could be detected using LAMP using gel electrophoresis (Figure 5B), SYBR™ Green I nucleic acid gel staining (Figure 5C) and real-time amplification with Genie^®^ III (Figure 5D) indicating that these are more sensitive than the detection limit of PCR. The detection limit of qPCR was 1.0 × 10^−4^ ng/µL (0.1 pg/µL), where cycle threshold value (Ct) was 34.98 ± 0.25 (Figure 5E). Therefore, the LAMP assay was more sensitive than PCR, but qPCR was the most sensitive assay among the three assays tested here.

### 2.5. Diagnosis of CuLCrV Infected Field Samples

The comparative effectiveness of the LAMP assay for detection of CuLCrV symptomatic and asymptomatic field samples was tested with qPCR and PCR assays (Table 1, Figure 6, Appendix A). All 10 symptomatic cucumber samples were found to be positive for CuLCrV by LAMP and qPCR; however, only 7 of these samples tested positive using PCR (Table 1). For the 10 asymptomatic cucumber samples, 7 tested positive by qPCR, 5 tested positive by LAMP, and only 2 tested positive using PCR (Table 1). Similarly, among 20 symptomatic winter squash samples, all of the samples tested positive via qPCR and 19 samples tested positive for CuLCrV via LAMP. In contrast, 15 samples exhibited positive amplification for CuLCrV using PCR (Table 1). For 20 asymptomatic winter squash samples, 11 samples were amplified using LAMP and qPCR, whereas, only 6 samples were positive for CuLCrV using PCR (Table 1). These results indicate that the LAMP assay was more sensitive than PCR for detection of infection with CuLCrV. However, qPCR was the most sensitive assay to detect infection within the field samples tested here (Table 1). Along with PCR and LAMP assays, quantitation of the virus titer by qPCR provided consistent findings, whereas the samples with lower amounts of virus showed smears or no bands for LAMP and PCR (Figure 6).

### 2.6. LAMP Amplification of Mixed Virus Infected Field Samples

In addition to the field samples tested above, seven new symptomatic winter squash leaf samples simultaneously collected from Lowndes County were tested for the presence of other begomoviruses and whitefly-transmitted viruses that are commonly found in Georgia [10]. Among eight different viruses that were tested, including CuLCrV, two other criniviruses, i.e., *Cucurbit yellow stunting disorder virus* (CYSDV) and *Cucurbit chlorotic yellows virus* (CCYV), were amplified using PCR with virus-specific primers (Figure 7A). To observe if the LAMP primer could amplify CuLCrV from the mixed infected samples, all those samples were tested for LAMP amplification. Successful LAMP amplified product was obtained in less than 20 min in the Genie^®^ III real-time amplification system (Figure 7B). Gel electrophoresis data also showed the LAMP amplified product with a characteristic ladder-like band pattern (Figure 7C). These data supported the fact that the LAMP can also amplify CuLCrV in mixed infected field samples.

## 3. Discussion

Whitefly-transmitted CuLCrV poses a threat to cucurbit production in the southeastern United States and causes enormous financial losses [7,10]. Therefore, early diagnosis of this virus is crucial to prevent further loss. Currently, the preferred method to differentiate the identity of the virus from other begomoviruses is PCR [3,5,6,12]. However, PCR is not sensitive enough compared to other molecular methods, is laborious and time-consuming, and requires a well-equipped laboratory, which makes this assay unavailable for detecting pathogens in field conditions [52,53]. The LAMP assay described here is a more rapid, accurate, sensitive, simple, and portable diagnosis method, which can be utilized in laboratory and field conditions for timely detection of this virus. 

To the best of our knowledge, this study is the first report of a LAMP assay for the detection of CuLCrV. In this study, the LAMP diagnostic method was established for the detection of CuLCrV from infected leaf samples. A set of six primers targeting the DNA-A gene was able to amplify the CuLCrV DNA-A gene. Optimization of the LAMP assay was performed using DNA extracted from CuLCrV infected squash as templates. The result revealed that the optimized reaction temperature was 71 °C for 60 min. The LAMP assay could detect virus infection from infected samples in as little as 15 min (Figure 6), whereas PCR assay takes at least 2.5 to 3 h for detection of the virus. In addition, the sensitivity of the LAMP assay was 10 times higher than PCR (Figure 5) and could detect the virus from symptomatic and asymptomatic plants with a greater efficiency compared to PCR (Table 1, Figure 6, Appendix A). Although compared to LAMP, qPCR was more sensitive for detection of virus infection (Figure 5, Table 1), it is not applicable for field detection of infection as the assay requires an expensive thermocycler and expert technicians [53]. Furthermore, the CuLCrV-LAMP assay was highly specific for detection of CuLCrV, as no amplification was observed from other closely related begmoviruses like SLCV (where the pairwise nucleotide identity is 81.8% and pairwise amino acid identity is 83.6% for DNA A sequences, Appendix A) in a mixed virus assay or mixed infected field samples or in healthy squash leaf samples (Figure 3, Figure 4 and Figure 7). The nucleotide alignment for LAMP primer binding sites of CuLCrV DNA A with the other 5 begomovirus DNA A sequences (Figure 8) provide the reason behind the specificity of the LAMP primers to amplify only CuLCrV. Where the number of single nucleotide polymorphisms (SNPs) present throughout all the six primer-binding sites ranged from 39 to 52 bp (Figure 8, Appendix A) including insertion and deletion mutations for some of the begomoviruses (MCLCV, *Squash yellow mild mottle virus* (SYMMV) and SLCV, Figure 8; Appendix A) for CuLCrV-LAMP primer binding sites. This information further supports the specificity of the assay to detect CuLCrV. 

This report supports results from other studies which indicate that the LAMP assay is broadly functional in plant pathogen detection [38,54,55], as it is a quite simple, rapid, sensitive, specific diagnostic method, requires only a water bath or heat block for incubation under isothermal conditions, and is adjustable to a range of detection approaches and settings [27,56]. The amplified products could be visualized using SYBR^®^ Green I nucleic acid gel staining, gel electrophoresis or real-time amplification by Genie^®^ III (Figure 2, Figure 3 and Figure 4) as in previous reports [36,57]. Taken together, these results suggest that LAMP is a good substitute for conventional PCR and other PCR-based methods like qPCR due to its simplicity and rapidity without the need of sophisticated instruments for on-site detection of pathogens [52,53]. In conclusion, the LAMP assay developed in this study is an efficient, reliable, and sensitive method for rapid detection of CuLCrV. Therefore, this method shows excellent potential as a valuable diagnostic tool that could be utilized for disease surveillance in infected cucurbits to detect CuLCrV infection when visible symptoms are present or in latent infections to prevent virus spread and further disease outbreaks.

## 4. Materials and Methods

### 4.1. Plant Materials

CuLCrV-infected leaves were sampled from commercial fields of vegetables including yellow squash, zucchini, watermelon and cucumber in Tift County, Georgia, USA and stored at -80°C for subsequent analysis. Other begomovirus-infected samples including SLCV and SMLCV -infected squash, TYLCV and one tospovirus TSWV -infected tomato were collected by vegetable laboratories in USDA-ARS, Charleston, SC and the Department of Plant Pathology at University of Georgia.

### 4.2. DNA and RNA Extraction

Total DNA was extracted from the DNA virus-infected plant samples including CuLCrV, SLCV, SMLCV, TYLCV using CTAB DNA extraction solution (G-Bioscience, St. Louis, Mo, USA) with a slight modification to the kit protocol. Briefly, infected leaf samples were flash-frozen in liquid nitrogen and pulverized using a mortar and pestle. About 100 mg of sample was resuspended in 500 μL of CTAB extraction solution containing 1% PVP and incubated at 65 °C for 20 min. Then, the samples were centrifuged at 12,000 x g for 5 min. Following RNase treatment and incubation at room temperature for 15 min, samples were centrifuged for an additional 5 min at 12,000 x g. After extracting the lysate with an equal volume of chloroform: isoamyl alcohol (24:1), samples were centrifuged for 5 min at 12,000 x g to separate the phases. Following the addition of 0.7 volumes of cold isopropanol to the transferred clear upper phase, samples were incubated at -20 °C for 10 min and then centrifuged (12,000 × g, 10 min). After decanting the supernatant, the pellet was washed with 500 μL cold 70% ethanol, and nucleic acids were recovered (12,000 × g, 5 min), resuspended in TE buffer (10 mM Tris–HCl, 0.5 mM EDTA, pH 8), and either used immediately or stored at −20 °C. 

For the detection and quantification of RNA viruses including TSWV, CYSDV, SqVYV, CCYV and ToCV, total RNA was extracted using RNeasy^®^ Plant Mini Kit (Qiagen, MD, USA) with a slight modification of the kit protocol. Total DNA and RNA yield and purity were estimated by measuring OD 260 nm and OD 260 nm/280 nm with a NanoDrop spectrophotometer (NANODROP LITE, Thermo Scientific, Waltham, MA, USA). Total RNA (1 µg) was converted to cDNA using iScript cDNA synthesis kit (Bio-Rad Laboratories, Hercules, CA, USA) according to the manufacturer’s protocol for subsequent use.

### 4.3. LAMP Primer Design and Comparison of CuLCrV DNA A Sequence with Other Begomovirus DNA Sequences

The published CuLCrV DNA-A complete sequence (GenBank accession number: NC_002984) was used to design LAMP primers using Primer Explorer version 5 (http://primerexplorer.jp/lampv5e/). The primers included two outer primers (F3 and B3), two inner primers (FIP and BIP), and two loop primers (LF, LB) (Figure 1, Appendix A). Primers were synthesized by Sigma-Aldrich (Sigma-Aldrich, St. Louis, MO, USA), dissolved in qPCR-grade water (Sigma-Aldrich) to produce 100 µM solutions, and stored at −20 °C. To check if the primer binding sites have any diverse sequence or insertion or deletion mutation with other closely related viruses, other begomoviruses sequences were also obtained from GenBank and used for comparison with CuLCrV sequence (Figure 8, Appendix A). These sequences were obtained from GenBank for DNA-A complete sequences, and their corresponding GenBank accession numbers are BCMV: accession number AF110189, SMLCV: accession number: DQ285014, SLCV: accession number: M38183.1, MCLCV: accession number: AF325497 and SYMMV: accession number: AY064391. The selected sequences were imported into Geneious v10.1.2 (Biomatters Ltd., Auckland, New Zealand) for sequence alignment using the Align/Assemble > Pairwise/Multiple Align function using “Geneious Alignment” option with default settings (Figure 8). To generate the alignment, the multiple alignment algorithm was used and then curated by hand. A pairwise identity between nucleotide sequences of CuLCrV DNA-A and the other 5 closely related begomovirus DNA-A sequences was recorded (Appendix A). For all sequences, open reading frames were identified and translated for subsequent protein alignments to check the pairwise identity between amino acid sequences. Instances of sequence divergence presented as single nucleotide polymorphism (SNP) and insertion/deletion mutations between the CuLCrV-LAMP primer binding sites, and the other begomovirus sequences were also recorded (Appendix A).

### 4.4. Optimization of LAMP Conditions to Detect CuLCrV Infection

To determine the optimal reaction conditions for detection of CuLCrV genomic DNA using LAMP, total DNA extracted from CuLCrV-infected squash leaf sample was used as a template. To determine the optimal reaction temperature, LAMP was performed from 62 to 69 °C (Appendix A) using 1.0 ng/µL and 66 to 73 °C using 1.0 × 10−1 ng/µL (Appendix A, Appendix A) of CuLCrV-infected DNA sample diluted from the initially extracted DNA sample. Uninfected healthy squash leaf samples and the reaction mix with molecular grade water were used as a healthy and negative control. The reaction was carried out using LavaLAMP™ DNA Master Mix (Lucigen, Middleton, WI, USA) according to the reaction mixture and protocol described in the method section. For real-time amplification, Genie^®^ III (OptiGene, Horsham, WS, UK) was used. Two main parameters were recorded to assess the amplification effectivity of the sample at the above-stated temperature range using the Genie^®^ III system: amplification time (Tiamp) and amplicon annealing temperature (Ta). When the fluorescence of the second derivative of the signal reaches its peak above the baseline value it is recorded as the Tiamp (expressed in minutes and seconds). On the other hand, Ta is the temperature (expressed in °C) at which double-stranded DNA product dissociates into single strands. To check the reliability of the Genie^®^ III system, amplified the product was checked using agarose gel electrophoresis and visual observation was performed with SYBR™ Green I (Invitrogen, Carlsbad, CA, USA).

To determine the optimum time for amplification, the LAMP reaction was carried out at 71 °C for 30, 45, 60 or 75 min intervals with 5.0 × 10−4 ng/µL of CuLCrV-infected DNA sample. The time required for each real-time amplification for the diluted sample was obtained from the Genie^®^ III LAMP and the optimal time was determined based on the results.

### 4.5. Reaction Conditions of LAMP, PCR, and qPCR

For comparative sensitivity analysis of LAMP with other molecular methods including PCR and qPCR, a series of 5-fold dilutions (10 to 10−5 ng/μL) of DNA extracts from CuLCrV-infected squash leaf samples were used as the templates under the optimized conditions. For LAMP assay, Genie^®^ III and Genie explorer software (OptiGene, Horsham, WS, UK) were used for real-time amplification and analyzing data, respectively. Genie^®^ III amplified products were further analyzed on 1% agarose gel electrophoresis for visual inspection and with SYBR™ Green 1 nucleic acid gel stain (Invitrogen, Carlsbad, CA, USA) for naked-eye observation. The reaction was performed with LavaLAMP™ DNA Master Mix (Lucigen, Middleton, WI, USA) in a 25 μL mixture containing 1.6 μM each of FIP and BIP; 0.2 μM each of F3 and B3; 0.8 μM each of loop F and loop B, respectively (Table 1); 12.5 μL 2× LavaLAMP™ DNA Master Mix, 1 μL Green Fluorescent Dye (Lucigen, Middleton, WI, USA), 1 μL DNA or cDNA template and nuclease-free H_2_O was added up to the volume. The mixture was preheated at 90 °C for 3 min, amplified at 71 °C for 60 min, and then terminated at a range from 98 to 80 °C, with a decline rate of 0.05 °C per second. 

PCR reactions were performed on a thermocycler (Biorad-96 well T100™, Bio-rad, Hercules, CA) using EconoTaq PLUS GREEN 2X Master Mix (Lucigen, Middleton, WI, USA) according to the manufacturer’s suggested protocol. For each PCR reaction, 1 μL of DNA or cDNA was added with 0.3 µM of each target-specific forward and reverse primers (Appendix A), 10 µL of 2X EconoTaq PLUS GREEN 2X Master Mix, and nuclease-free H_2_O was added up to a final volume of 20 µL. For the sensitivity analyses of the CuLCrV PCR primers, 1 μL of the serially-diluted DNA was used. For all other PCR, 1 µL of DNA or cDNA was used with virus-specific primers designed in this study or obtained from previous studies with the reaction protocol described in Appendix A. Amplified products were detected on a 1% agarose gel electrophoresis following staining with GelGreen^®^ Nucleic Acid Gel Stain (Biotium, Fremont, CA, USA) and visualization on a UV transilluminator UVP UVsolo touch (Analytik Jena, Upland, CA, USA). 

The qPCR assay was performed in a PikoReal™ Real-Time PCR System (Thermofisher Scientific, Waltham, MA, USA) using iQ™ SYBR Green Supermix (BioRad Laboratories Inc., Hercules, CA, USA) in a 10 µL reaction using primers designed in this study or obtained from other studies (Appendix A) according to the manufacturer’s protocol. Each reaction mixture contained 5 µL of Biorad iQ™ SYBR Green Supermix, 0.3 µM of each of the target-specific forward and reverse primers (Appendix A), 1 µL of DNA or cDNA sample, and nuclease-free H2O was added up to a volume of 10 µL. Optimal thermocycling conditions were used for all reactions beginning with an initial denaturing step of 95 °C for 120 s with optics off, followed by 40 cycles at 95 °C for 10 s with optics off, and 60 °C for 50 s with optics on and a temperature ramp from 60 °C to 95 °C at 0.2 °C/s for melting curve analysis (Appendix A). For comparative sensitivity analysis, 1 μL of the serially-diluted DNA was used. For all other qPCR assays, 1 µL of DNA or cDNA was used with the target-specific primers and reaction protocol described in Appendix A. Samples were amplified in triplicate including one positive control from the DNA extract of CuLCrV infected squash leaf, one healthy control from healthy squash leaf DNA, and deionized water as negative control. The cycle threshold (Ct) value was obtained from PikoReal™ Software 2.1 (Thermofisher Scientific, Waltham, MA, USA) after the amplification was done.

### 4.6. Specificity Analysis of LAMP

The specificity of LAMP primers for the detection of CuLCrV was examined using 50 ng of total DNA extracted from CuLCrV-infected squash leaf and other begomoviruses, i.e., SLCV- and SMLCV-infected squash, TYLCV and cDNA of one tospovirus TSWV-infected tomato leaves as the templates. Quantification of the virus titer was performed using qPCR relative to the internal control gene Rubisco. The DNA and RNA extraction from leaf tissues and LAMP assay were performed according to the methods described above.

To further test the specificity of this LAMP assay for the detection of CuLCrV in a mixed virus assay, five different template mixtures were prepared: 1. CuLCrV with other begomovirus DNA mixture: 2 µL each of the DNA from CuLCrV and two other begomoviruses, i.e., SLCV and TYLCV were added and mixed in one 1.5 mL Eppendorf tube. 2. CuLCrV with healthy control DNA: 2 µL of DNA from CuLCrV was mixed with 4 µL of DNA extracted from healthy squash leaf sample in another tube. 3. For control studies, one control was with 2 µL of DNA from each of the virus, i.e., SLCV and TYLCV with 2 µL DNA of healthy squash leaf sample. 4. Other controls were with DNA from a healthy squash leaf sample as the healthy control and 5. deionized water as method control. For LAMP and PCR assays, 1µL from each of the mixed DNA samples were used as a template. CuLCrV specific primers were used to do PCR and subsequent direct sequencing of the PCR products (Retrogen Inc., San Diego, CA, USA) for further confirmation of the detection of CuLCrV. 

### 4.7. Evaluation of LAMP with CuLCrV-Infected Field Samples

CuLCrV infected cucumber and winter squash leaf samples grown in several fields from Tift and Lowndes counties in Georgia, USA, were collected for LAMP analysis (Appendix A). Twenty cucumber leaf samples and forty winter squash leaf samples from both symptomatic and asymptomatic plants collected from the two counties were tested using qPCR, PCR and LAMP methods followed by DNA extraction according to the manufacturer’s protocol. Healthy squash leaf tissues were used as the healthy control and deionized water as a negative control. For each assay, 50 ng of DNA was used. All amplification was done using Genie^®^ III (OptiGene, Horsham, WS, UK) for real-time LAMP amplification. For further confirmation, the products were separated and analyzed on a 1.0% agarose gel and visualized by adding SYBR™ Green 1 nucleic acid gel stain, and the positive and negative results obtained from each method were compared and recorded in a table. Three representative cucumber and winter squash leaf samples from both symptomatic and asymptomatic plants were visualized (Figure 6) to show the comparative sensitivity between PCR and LAMP. 

Apart from the field samples that were tested above, seven more symptomatic infected winter squash leaf samples collected from Lowndes county were tested to check whether the field samples were mixed virus-infected or not, and to check if the LAMP amplification will be inhibited because of the mixed virus infection. To carry out the experiment, homogenized leaf tissues were used for both RNA and DNA extraction followed by the method described above. DNA was used as a template for PCR to test the presence of DNA viruses, i.e., CuLCrV, SLCV, SMLCV and TYLCV and cDNA was used for the RNA viruses, i.e., CYSDV, SqVYV, CCYV and ToCV with the virus-specific primers (Appendix A).

## Figures and Tables

**Figure 1 ijms-21-01756-f001:**
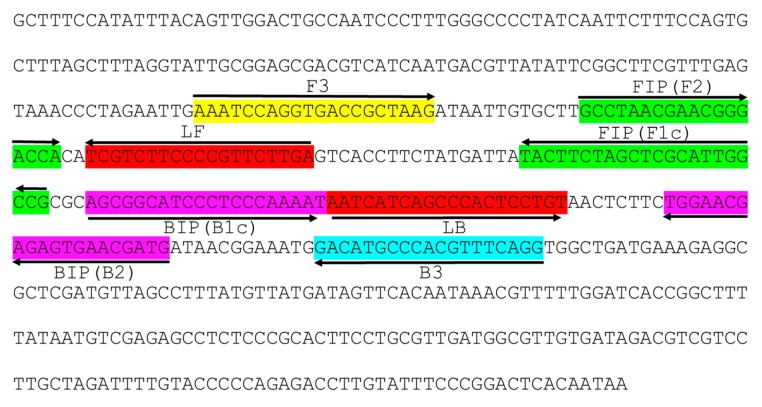
Location and partial sequence of loop-mediated isothermal amplification (LAMP) primer sets targeting *Cucurbit leaf crumple virus* (CuLCrV) -specific DNA. Locations for two outer (F3 and B3), two inner (FIP [F1c-F2], and BIP [B1c-B2]) and two looping (LF and LB) primers are indicated in the figure with colors. FIP is a hybrid primer consisting of the F1c sequence and the F2 sequence, and BIP is a hybrid primer consisting of the B1c sequence and the B2 sequence. Arrows indicate the extension direction. Here, we used yellow color for F3, sky blue color for B3, green for FIP, pink for BIP and red for two loop primers LF and LB.

**Figure 2 ijms-21-01756-f002:**
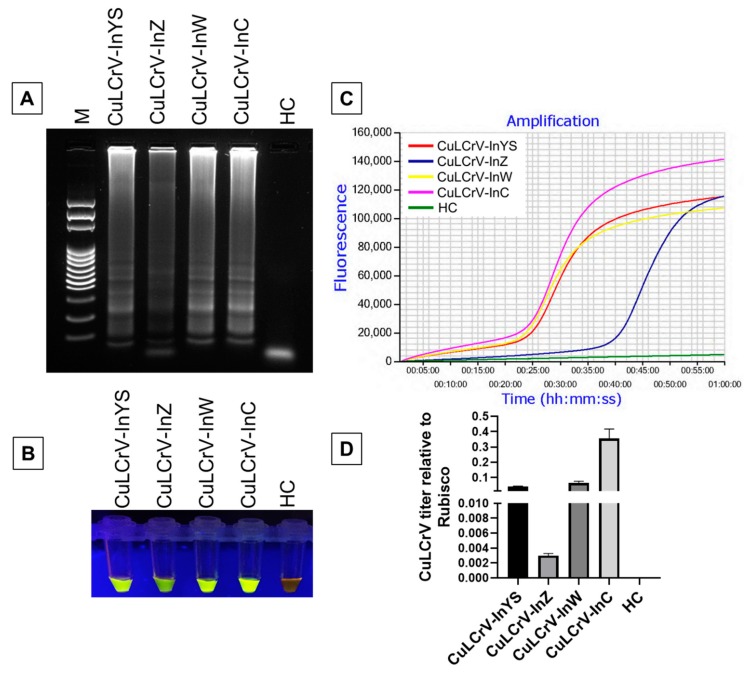
Detection of CuLCrV from CuLCrV-infected plants of squash, zucchini, watermelon and cucumber by LAMP. Successful amplification was visualized using the agarose gel image (**A**), the naked colorimetric visual inspection with SYBR green 1 DNA gel staining method (**B**), and real-time amplification using Genie III (**C**). The CuLCrV titer was calculated by qPCR relative to internal control gene Rubisco (**D**). Here, we used CuLCrV-InYS: CuLCrV-infected yellow squash, CuLCrV-InZ: CuLCrV-infected zucchini, CuLCrV-InW: CuLCrV-infected watermelon, CuLCrV-InC: CuLCrV-infected cucumber, lane M: 100 bp DNA ladder, and lane HC: healthy squash leaf as the negative control (**A**). The fluorescence green-colored products could be visualized after SYBR green 1 DNA gel staining under UV light, except negative reaction (**B**).

**Figure 3 ijms-21-01756-f003:**
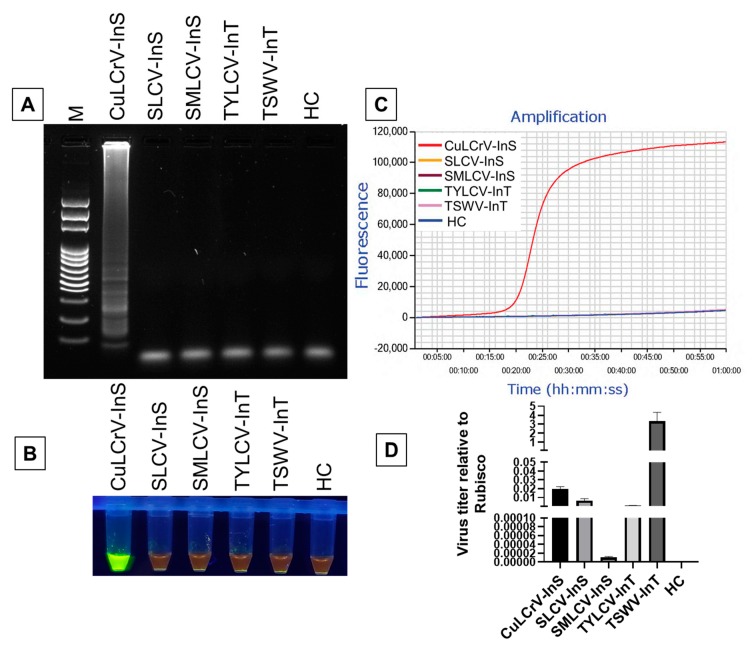
The specificity of the LAMP assay for the detection of CuLCrV. Amplified products were analyzed using agarose gel electrophoresis (**A**), visual inspection with SYBR™ green 1 DNA gel staining (**B**), and real-time amplification using Genie^®^ III (**C**). The virus titer was calculated using qPCR relative to internal control gene Rubisco (**D**). We used CuLCrV-InS: CuLCrV-infected squash, SLCV-InS: SLCV-infected squash, SMLCV-InS: SMLCV-infected squash, TYLCV-InT: TYLCV-infected tomato, TSWV-InT: TSWV-infected tomato, HC: healthy squash leaf as a negative control, and M: 100 bp DNA ladder (**A**). The fluorescent green-colored products could be visualized after SYBR™ green 1 DNA gel staining under UV light while negative reactions remained orange in color (**B**). Successful LAMP amplification is demonstrated by curve amplified by Genie^®^ III (C).

**Figure 4 ijms-21-01756-f004:**
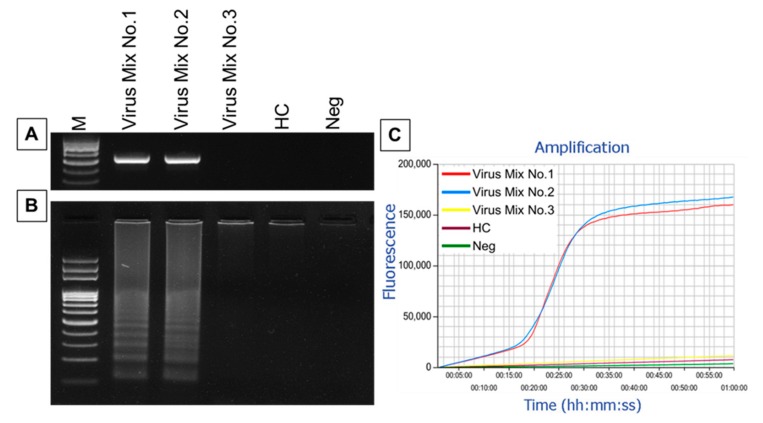
Specificity of the LAMP assay for the detection of CuLCrV in the mix virus assay. CuLCrV was detected from mixed virus DNA with virus-specific primers using PCR (**A**). LAMP amplified products were analyzed using agarose gel electrophoresis (**B**), and real-time amplification with Genie^®^ III (**C**). Here, we used Virus Mix No. 1: CuLCrV with other viruses including SLCV and TYLCV, Virus Mix No. 2: CuLCrV with the healthy control, Virus Mix No. 3: mix viruses, i.e., SLCV and TYLCV excluding CuLCrV with healthy control, HC: healthy squash leaf as the healthy control, Neg: DH_2_0 as the negative control, and M: 100 bp DNA ladder (**A**,**B**). Successful LAMP amplification is demonstrated by the curve amplified using Genie^®^ III (**C**).

**Figure 5 ijms-21-01756-f005:**
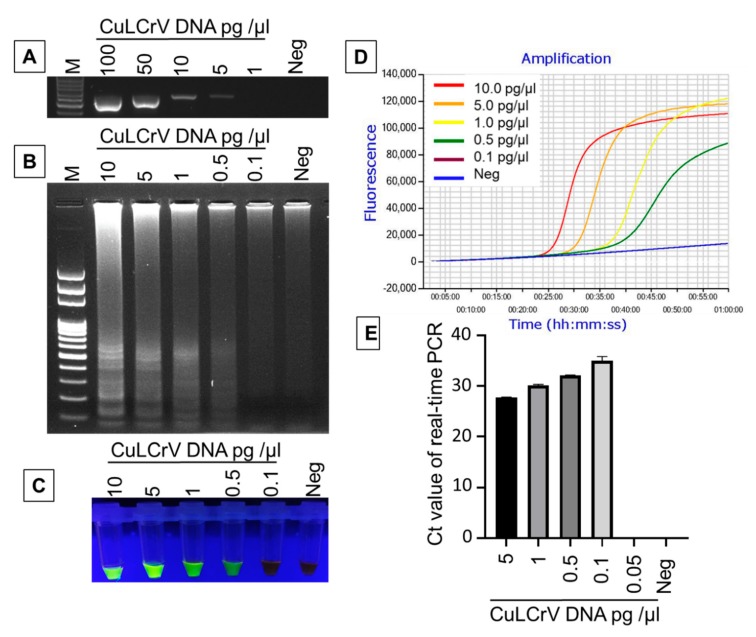
Comparative sensitivity analysis of LAMP (**B**–**D**), PCR (**A**), and qPCR (**E**) for the detection of CuLCrV. Amplified PCR products were analyzed using agarose gel electrophoresis (**A**), whereas, the LAMP product was analyzed using agarose gel electrophoresis (**B**), SYBR™ Green I nucleic acid gel staining (**C**), and Genie^®^ III (**D**). M: 100 bp ladder marker (**A**,**B**), Neg: negative control (**A**–**E**); others are serially diluted DNA concentrations in picogram per microliter (pg/µL) (**A**–**E**). The fluorescence green-colored products could be visualized after SYBR^®^ green 1 DNA gel staining under UV light for the amplified LAMP product, while negative reactions produced an orange color (**B**). Real-time LAMP amplification was done using Genie^®^ III (**D**). DNA concentrations are in picogram per microliter (pg/µL) from the serially diluted DNA extract from the CuLCrV-infected squash sample.

**Figure 6 ijms-21-01756-f006:**
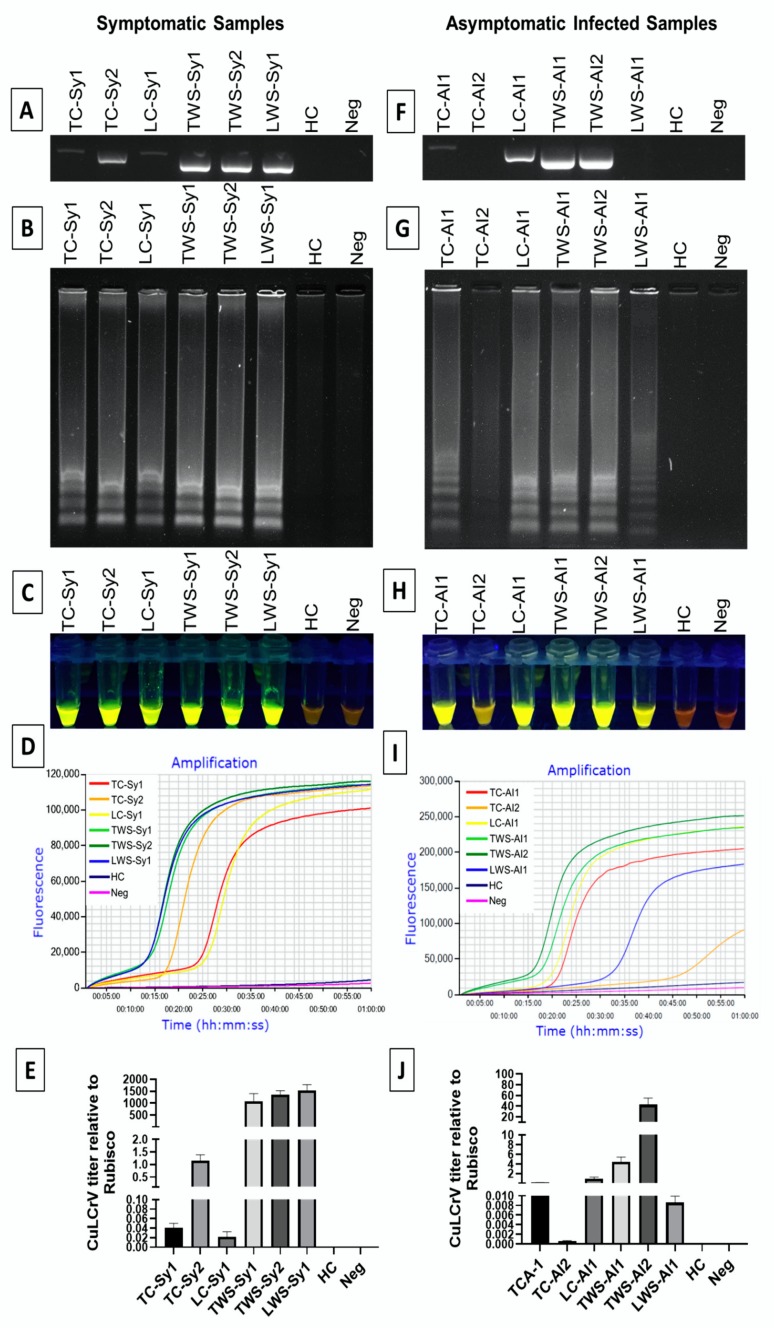
Detection of CuLCrV from symptomatic (**A**–**E**) and asymptomatic infected (**F**,**G**,**H**,**I**,**J**) cucumber and winter squash leaf samples using LAMP (**B**–**D**,**G**–**I**), PCR (**A**,**F**) and qPCR (**E**,**J**). LAMP amplified products were analyzed using agarose gel electrophoresis (**B**,**G**), and visual inspection with SYBR™ green 1 DNA gel staining, where positive amplification gave a fluorescent green color and negative reactions remained orange under UV light (**C**,**H**), and real-time amplification with Genie^®^ III (**D**,**I**). CuLCrV titer was quantified using qPCR relative to internal control gene Rubisco (**E**,**J**). Here, we used TC-Sy1, 2: Tift county cucumber symptomatic infected leaf sample no. 1,2; LC-Sy1: Lowndes county cucumber symptomatic infected sample no. 1; TWS-Sy1, 2: Tift county winter squash symptomatic infected leaf sample no. 1,2; LWS-Sy1: Lowndes county winter squash symptomatic infected sample no. 1; TC-AI1, 2: Tift county cucumber asymptomatic infected leaf sample no. 1,2; LC-AI1: Lowndes county cucumber asymptomatic infected sample no. 1; TWS-AI1, 2: Tift county winter squash asymptomatic infected leaf sample no. 1,2; LWS-AI1: Lowndes county winter squash asymptomatic infected sample no. 1; HC: healthy control, Neg: negative control or non-template control.

**Figure 7 ijms-21-01756-f007:**
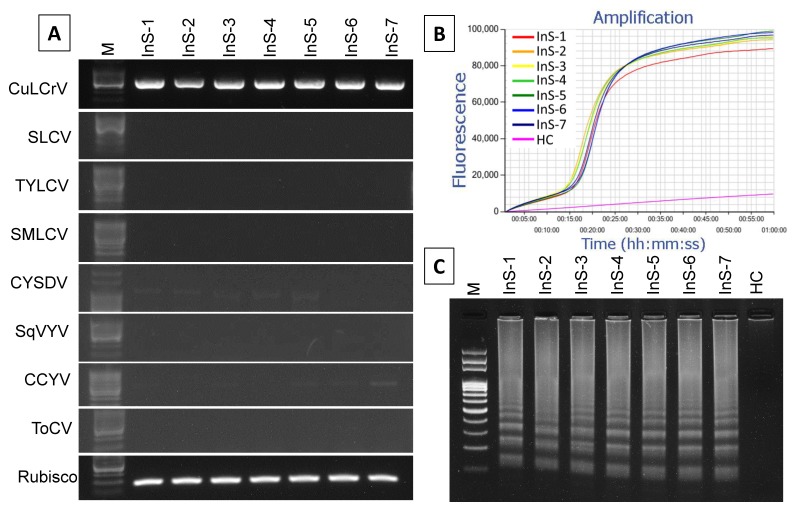
LAMP amplification of CuLCrV in mix infected winter squash field samples. Virus-specific primers by PCR (CuLCrV, SLCV, TYLCV and SMLCV) and RT-PCR (CYSDV- *Cucurbit yellow stunting disorder virus*, SqVYV- *Squash vein yellowing virus*, CCYV- *Cucurbit chlorotic yellows virus* and ToCV- *Tomato chlorosis virus*) detected infections of several viruses from infected winter squash collected from Lowndes county (**A**). Rubisco was amplified as an internal control gene (**A**). LAMP amplified products were analyzed by agarose gel electrophoresis (**C**), and real-time amplification by Genie^®^ III (**B**). Here, InS-1 to 7: infected samples 1 to 7 collected from field, HC: healthy squash leaf as healthy control, and M: 100 bp DNA ladder (**A**,**C**). Successful LAMP amplification is demonstrated by the curve amplified by Genie^®^ III (**C**).

**Figure 8 ijms-21-01756-f008:**
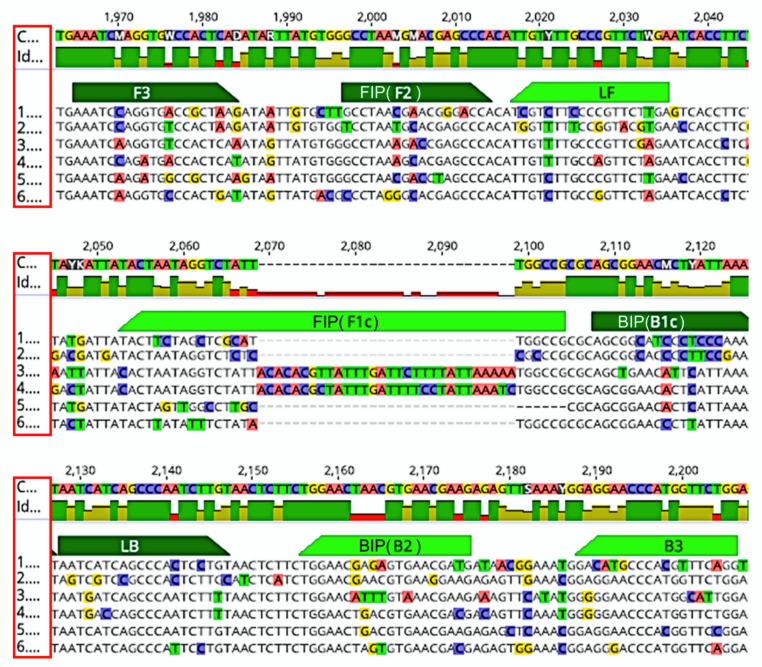
The specificity of six LAMP primers (F3, B3, FIP [F1c-F2], and BIP [B1c-B2]) to bind to the CuLCrV-specific gene sequence (DNA-A). An alignment shows single-nucleotide polymorphisms (SNPs), insertion and deletion mutations (colored letters) for CuLCrV LAMP primer binding sites to other begomoviruses. The red box is showing the numbers for the virus sequences, where 1: CuLCrV, 2: BCMV, 3: MCLCV, 4: SYMMV, 5: SLCV and 6: SMLCV. Different colors within nucleotides are showing SNPs and insertion /deletion mutations among viruses.

**Table 1 ijms-21-01756-t001:** LAMP detection of CuLCrV from symptomatic and asymptomatic leaf samples collected from fields in Georgia, USA.

Sample	Geographic Location	Symptomatic/Asymptomatic	No. of Samples Tested	Detected by qPCR	Detected by LAMP	Detected by PCR
Cucumber	Tift County	Symptomatic	5	5	5	3
		Asymptomatic	5	4	3	1
	Lowndes County	Symptomatic	5	5	5	4
		Asymptomatic	5	3	2	1
Winter Squash	Tift County	Symptomatic	10	10	10	8
		Asymptomatic	10	6	6	3
	Lowndes County	Symptomatic	10	10	9	7
		Asymptomatic	10	5	5	3

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
