# Peer review of "Development of Loop-Mediated Isothermal Amplification Assay for Rapid Detection of Cucurbit Leaf Crumple Virus"

_ijms, 2020, doi:10.3390/ijms21051756_

Round 1

Reviewer 1 Report

The manuscript of Waliullah et al. describes an LAMP-based assay for specific detection of CuLCrV in detail, not lengthily. Bottom line is most importantly are the primers for LAMP. So according to Fig. 1 arrow of FIP is forward. Table 1 says it is reverse, as in Fig. 6. Please double-check and correct Figs or Table. These primers need to be analyzed thoroughly, which was done by the authors, except some things I really want to be clarified. First, specificity of the primers for CuLCrV was tested against SLCV, SMLCV, TYLCV and TSWV, but not in a mixed infection assay, correct? In a mixed infected field sample the different DNA components would be all present in one sample. An additional important assay is to mix all tested DNAs and analyze if only CuLCrV DNA will be amplified, i.e. by PCR, cloning & sequencing or direct sequencing. I’m asking this because the field samples in the assay shown in Fig 5 are not mixed infected, correct? Have these plants been tested by a standard routine assay in addition, i.e. ELISA? Is it known what is in these samples? If you validate your method then all samples should have been tested by the golden standard in addition. It is not clear which samples from Table 2 are shown in Fig. 5. All have been tested, why not show all?

Minor comments:

l.87: remove RT from RT-LAMP

l.97: show the data in suppl.

l.109-114: check format, font height

l.187: don’t divide tables by page

l.199: use abbr. once it was introduced, delete cucurbit leaf crumple virus

Fig. 5: It is not useful to use the same name for two different samples, i.e. Cu-1 (A) and Cu-1 (E). Indicate in the figure what was symptomatic and asymptomatic, maybe combine with pics from suppl. Fig 2.

Fig. 6, legend: where is the red box?

Author Response

Dear Editor,

Please see the attached file for our responses to your comments. 

Thank you,

Emran Ali

Reviewer 2 Report

The manuscript cannot be accepted in its current form because of an apparently confused first part and because it forgets to continue the comparison with qPCR and omits to consider the viral titer in the samples.

List of suggestions:

Bemisia tabaci must be written in Italics “Bemisia tabaci”, the same for Bst at line 72 being the abbreviation of Bacillus stearothermophilus. Lines 100-101: Please explain how “The peak amplification time of 46 min 15 sec was obtained from 60 and 75 min reaction cycles”. Line 106: please explain much better the reasons of the optimal reaction condition of 71° C and 60 min. Figure 1 and Table 1 do not fit together because, if the orientation of the primers is correct in the figure 1, the nucleotide sequences in Table 1 are not correct (at least some of them); this begs the question of which primers were used. Figure 2: a qPCR would add value to the figure. Figure 3: other viruses are not detected but this can simply result from a low virus titer; please add virus titer and / or qPCR; also, it is better to use the terms “infected symptomatic” and “infected asymptomatic” Table 2 / Figure 5: the virus titer or a comparative qPCR should be added; furthermore, how many of the asymptomatic samples turn out to be infected? How was this test done? Figure 5: please give a different name to symptomatic samples than asymptomatic samples. A list of abbreviations should be included in the MS.

Author Response

Dear Reviewer, 

Please see the attached file for our responses to your comments. 

Thank you so much,

Emran Ali

Round 2

Reviewer 2 Report

The MS is really improved but it is difficult to read because of too many acronyms of not immediate comprehension as well as rather complicated figure legends.

Therefore, I suggest:

  • An English check is necessary (e.g. see abstract lines 19-21: “The sensitivity assay revealed that LAMP was more sensitive than conventional PCR, but less sensitive than qPCR.””;
  • LAMP primer sets – Figure 1, Supplementary Table 1 – lines 11-113: in the text the primers are six namely F3, B3, FIP, BIP, LF and LB, in the Figure 1 they seem to be eight because of the existence of BIP(B1c), BIP(B2), FIP(F2) and FIP(F1c) but in the Legend of Figure is written that “FIP is a hybrid primer consisting of the F1c sequence and the F2 sequence” and that “BIP is a hybrid primer consisting of the B1c sequence and the B2 sequence”; Because in the figure the couple of primers F1c and F2, B1c and B2 show between the couple an opposite orientation some problem arises; furthermore, in the Figure 8 Authors use eight acronyms for primers (F3, F2, LF, F1c, B1c, LB, B2, B3) which differ from Figure 1; consequently, a revision of this part is mandatory;
  • Figure 2 B should be positioned under the columns 2-6 of Figure 2 A to facilitate the comparison;
  • Figure 2 and Figure 3 acronyms: please use more explanatory acronyms example CuLCrV-InZ where InZ stays for Infected Zucchini or TSWV-InT where InT stays for Infected Tomato;
  • Figure 3 B should be positioned under the columns 2-7 of Figure 3 A to facilitate the comparison;
  • Figure 4: on the top of Figure 4A it is necessary to write “Virus mix no.”;
  • Figure 5: it is simpler to write on top of Figure 5A “CuLCrV DNA as femtogram for microliter” or “CuLCrV DNA fg /µl” to have shorter number on top of each Figure or below the x axis;,
  • Figure 5 C should be positioned under the columns 2-7 of Figure 5 B to facilitate the comparison;
  • Figure 6: it is simpler to write on top, in the left “symptomatic samples” and in the right “asymptomatic infected samples”; furthermore, the acronyms could derive from the plant name plus + abbreviation of symptomatic or asymptomatic infected, e.g TWS-Sy1, TWS-Sy2, or TWS-AI1, TWS-AI2;
  • Figure 7: on top of Figure 7A, 7C, is opportune to write “Infected sample no.” and within Figure 7B Sample 1 to 7;

Author Response

Dear reviewer,

Thank you for the comments and suggestions on our manuscript “ijms-706969". We have attempted to address all of your comments in the revised version of the manuscript. Please see the attached file for our responses to your comments. Changes to the manuscript are viewable within the revised manuscript by red color.

Thank you so much,

Emran Ali

University of Georgia, USA
